# OpenReview forum: "MATA: Memory-Augmented Temporal Anchors for Sparse-Time Dynamic Knowledge Graph Embedding"
_ICLR.cc/2026/Conference — Submitted to ICLR 2026_

### Official Review · Reviewer_nU5v · 2025-10-15

**Soundness:** 1
**Presentation:** 1
**Contribution:** 1
**Rating:** 2
**Confidence:** 5

**Summary:**

The paper introduces MATA (Memory-Augmented Temporal Anchors), a framework for temporal knowledge graph (TKG) embedding under sparse-time conditions. The method includes a learnable memory of “temporal anchors” to interpolate embeddings for missing timestamps, combined with a contrastive temporal consistency loss. Experiments are reported on four benchmark datasets (ICEWS14, GDELT, WikiKG, YAGO-T), where MATA, according to the authors, outperforms several earlier TKG models.

**Strengths:**

* The ablation study (Table 3) demonstrates that each proposed component contributes to performance.
* The notation table is helpful.
* it seems like the hyperparameters of the method are somewhat robust, and can be re-used for different datasets without hyperparameter tuning.

**Weaknesses:**

1. Unconvincing Motivation

The paper claims that “real-world TKGs are incomplete, timestamps are often missing, and semantic drift can throw models off course,” yet provides no real-world evidence, examples, or citations for these claims.
All experiments are conducted on standard, fully timestamped datasets (ICEWS14, GDELT, etc.), and temporal sparsity is introduced artificially by randomly removing up to 70 % of timestamps. This setting is not realistic and undermines the core motivation.
The authors need to show either a real dataset exhibiting this issue or a practical scenario where timestamp sparsity genuinely arises.

2. Experimental Rigor and Reproducibility
* The paper lacks key experimental details: filtering strategy (static vs. time-aware), single-vs-multi-step prediction, train/validation/test splits, hyperparameters, scoring function, and code availability are all missing. I recommend reading [1] for more information on evaluation of TKG forecasting.
* Dataset statistics and references are not provided. For example, “YAGO-T” is not clearly defined, and ICEWS18, a standard benchmark is omitted without explanation. [1] describes that there exist multiple versions of e.g. ICEWS14, thus is is important to name the version/dataset size etc of the datasets for reproducibility.
* Only MRR is reported; Hits@k metrics are absent (even though they are promised  on page 5).
* Baselines: the baselines are outdated (mostly ≤ 2021) and likely not comparable under the same settings (e.g. filter settings [1]  unclear, single-vs multi-step prediction unclear).

3. Outdated and Incomplete Baselines

The comparison excludes almost all recent TKG models. Regcn [2] , TiRGN [3], CognTKE [4], Tlogic [5], Recurrency Baseline [6] (as a baseline). The claim of “state-of-the-art” performance is therefore unjustified.  I expect at least the recurrency baseline would work well under the sparse temporal setting, and potentially also Tlogic, the rule-based approach.

4. Clarity and Structure
* The introduction is poorly structured: the task (temporal extrapolation vs. interpolation) is introduced too late, confusing the scope of the paper. Further, the introduction is lacking references (0 references).
* The related work section is incomplete and mixes problem formulations (interpolation, extrapolation, rule-based inference) without clear distinctions. For example all rule-based approaches (Tlogic, CognTKE, Infer,..) are completely missing. I recommend to write less information for each method, but try to structure it better, and have a more concise overview.
* "To our knowledge, MATA is the first TKG framework to apply temporal contrastive learning." -> no, constrastive learning for TKG was e.g. tackled by CENET [7]
* The method section is superficial. I would appreciate a short motivation for each component, and an explanation of how the sub-modules work together (or a little figure). Several key information is missing, e.g. how is L_link computed? Which scoring function is used? How are the node and relation embeddings initialized? As far as I understand, the only way how the graph structure is taken into account is via the scoring function, but it is not clear what scoring fct is used?
* Citations are inconsistent (wrong LaTeX usage: \cite vs. \citet/\citep) and many statements lack references altogether (e.g. no reference in introduction).

5. Evaluation Design and Validity

The simulated missing-timestamp experiments (random removal of 30–70 %) do not reflect any realistic data condition and seem tailored to favor the proposed interpolation mechanism.
Without real-world evidence of timestamp sparsity or irregularity, the empirical results have little practical significance, and thus the motivation and superiority of the proposed method is not credible.


## Overall Assessment
While the idea of combining learnable temporal anchors with contrastive learning is somewhat interesting, and the approach is reasonably simply, the paper fails to justify why this problem matters in real TKGs, and the experimental setup lacks rigor, comparability, and transparency. The motivation is not convincing, the baselines are outdated and incomplete, and the results cannot be trusted without reproducibility details. Writing and referencing also fall below the clarity standards of ICLR.
Mainly for Weaknesses 1-3 I recommend rejecting the paper.

## References
[1]  Gastinger, J., Sztyler, T., Sharma, L., Schuelke, A., & Stuckenschmidt, H. (2023, September). Comparing apples and oranges? on the evaluation of methods for temporal knowledge graph forecasting. In Joint European conference on machine learning and knowledge discovery in databases (pp. 533-549). Cham: Springer Nature Switzerland.

[2] Zixuan Li, Xiaolong Jin, Wei Li, Saiping Guan, Jiafeng Guo, Huawei Shen, Yuanzhuo Wang, and
Xueqi Cheng. Temporal knowledge graph reasoning based on evolutional representation learning.
In The 44th International ACM SIGIR Conference on Research and Development in Information
Retrieval (SIGIR), 2021b.

[3] Yujia Li, Shiliang Sun, and Jing Zhao. TiRGN: Time-guided recurrent graph network with local-
global historical patterns for temporal knowledge graph reasoning. In Proceedings of the 31st
International Joint Conference on Artificial Intelligence (IJCAI), pp. 2152–2158, 2022a.

[4]  Wei Chen, Yuting Wu, Shuhan Wu, Zhiyu Zhang, Mengqi Liao, Youfang Lin, and Huaiyu Wan.
Cogntke: A cognitive temporal knowledge extrapolation framework. In Proceedings of the AAAI
Conference on Artificial Intelligence, volume 39, pp. 14815–14823, 2025.

[5]  Yushan Liu, Yunpu Ma, Marcel Hildebrandt, Mitchell Joblin, and Volker Tresp. TLogic: Temporal
logical rules for explainable link forecasting on temporal knowledge graphs. In 36th Conference
on Artificial Intelligence (AAAI), pp. 4120–4127, 2022

[6] Julia Gastinger, Christian Meilicke, Federico Errica, Timo Sztyler, Anett Schuelke, and Heiner
Stuckenschmidt. History repeats itself: A baseline for temporal knowledge graph forecasting.
In Proceedings of the Thirty-Third International Joint Conference on Artificial Intelligence (IJ-
CAI), 2024b

[7]  Yi Xu, Junjie Ou, Hui Xu, and Luoyi Fu. Temporal knowledge graph reasoning with historical
contrastive learning. In 37th Conference on Artificial Intelligence (AAAI), pp. 4765–4773, 2023

**Questions:**

Q1: Can you please clarify fully the experimental setup? (single-vs multi-step, time-aware or static filter, dataset versions/splits/statistics, hyperparameters tested and used)

Q2: What scoring function did you use? Did you test different scoring functions? Do I understand correctly, that the only way how the graph structure is taken into account is via the scoring function?

Q3: Missing baseline comparisons: Could you integrate comparison to my above mentioned baselines?

Q4: Do you have a real-world use case/reference/ dataset for the missing timestep scenario?

---

### Official Review · Reviewer_5xTR · 2025-10-16

**Soundness:** 1
**Presentation:** 2
**Contribution:** 1
**Rating:** 2
**Confidence:** 3

**Summary:**

This paper proposes MATA to address the limitations of existing modelsunder sparse-time conditions. It introduces a learnable set of temporal anchors stored in a differentiable memory module, enabling attention-based interpolation of timestamp-aware embeddings even at unseen or missing timepoints.  Experiment on four standard TKG benchmarks demonstrate the performance of MATA.

**Strengths:**

1. MATA enables attention-based interpolation of timestamp-aware embeddings even at unseen or missing timepoints.
2. A cross-version alignment objective stabilizes entity and relation embeddings across evolving KG snapshots, enhancing long-term coherence.

**Weaknesses:**

1. It is not clear how the problem of data is incomplete, timestamps are often missing, and semantic drift, are addressed by this method, since these strategy are not novel.
2. The presentation is vague. For example, what is the scoring function in the inference procedure.

**Questions:**

1. What is the difference between CONTRASTIVE TEMPORAL CONSISTENCY and CROSS-VERSION EMBEDDING ALIGNMENT?
2. CROSS-VERSION means the graph in different timestamps?

---

### Official Review · Reviewer_PzwV · 2025-10-23

**Soundness:** 2
**Presentation:** 1
**Contribution:** 1
**Rating:** 0
**Confidence:** 5

**Summary:**

The MATA (Memory Augmented Temporal Anchors) framework proposed in this paper innovatively introduces a differentiable memory module to store learnable temporal anchors for the problem of embedding sparse temporal dynamic knowledge graphs. By combining the contrastive temporal consistency loss and cross version alignment objectives, it effectively solves the limitations of existing models in scenarios such as temporal sparsity and semantic drift.

**Strengths:**

The Temporal Anchor Memory module is ingeniously designed to embed and interpolate any timestamp (including unobserved timestamps) through attention mechanisms, breaking through the limitations of traditional models that rely on dense time sampling or fixed time encoding, and providing a new approach to solving time sparsity problems.

**Weaknesses:**

1. The description in the model section of this article is very unclear and not detailed.

2. The motivation of this article is unclear, and the research on related work is too outdated

**Questions:**

see above

---

### Official Review · Reviewer_z9VH · 2025-10-26

**Soundness:** 2
**Presentation:** 2
**Contribution:** 2
**Rating:** 2
**Confidence:** 4

**Summary:**

This paper introduces MATA, a framework for learning embeddings on Temporal Knowledge Graphs. The framework includes a learnable temporal anchor mechanism to capture historical information from different timestamps. On this basis, the authors use a contrastive temporal consistency loss to smooth the evolution of entity representations across nearby timestamps. Experimental results show the effectiveness of MATA.

**Strengths:**

S1. Learning representations for TKGs under the sparse time scenario is interesting.

**Weaknesses:**

W1. The proposed framework appears to lack technical contributions. Specifically, the design of learnable anchor embeddings seems relatively naive, and it remains unclear how they effectively address the sparsity issue. Moreover, the temporal consistency loss adopted in the work is a commonly used technique and does not introduce notable novelty.

W2. The paper lacks a comprehensive analysis of previous works and studies. It contains only a few citations.

**Questions:**

None

---

### Meta-Review · Area_Chair_Nrkw · 2026-01-12

**Summary:**

All reviewers express concerns regarding its novelty, unclear motivation, and insufficient model description. The research on related work is too outdated, the structure is poorly organized, and the experimental results cannot be trusted without reproducibility details.  Besides, the overall writing quality requires significant improvement.

**Reviewer Concerns:**

All concerns are not be addressed since the authors do not provide response. Especially, Reviewer z9VH holds that the proposed method lacks technical contributions, and the analysis of previous works and studies is not comprehensive. Reviewer PzwV holds that the model description is very unclear and not detailed, and the motivation is also unclear. Reviewer 5xTR holds that the presentation is vague, and the used strategies are not novel. Reviewer nU5v holds that the experimental setting is not realistic, and key experimental details are missing.

**Reviewer Scores:**

Since there is no response, all reviewers will maintain their original scores. In particular, Reviewer z9VH mainly concerns its limited technical contributions and narrow previous work analysis.  Reviewer PzwV mainly concerns its unclear motivation and model description. Reviewer 5xTR mainly concerns its vague presentation and trivial strategies. Reviewer nU5v mainly concerns its unrealistic experimental details and poor structure.

---

### Decision · Program_Chairs · 2026-01-26

Reject